# A KOOPMAN APPROACH TO UNDERSTANDING SEQUENCE NEURAL MODELS

## ABSTRACT

Deep learning models are often treated as "black boxes". Existing approaches for understanding the decision mechanisms of neural networks provide limited explanations or depend on local theories, such as fixed-point analysis. Recently, a data-driven framework based on Koopman theory was developed for the analysis of nonlinear dynamical systems. In this paper, we introduce a new approach to understanding trained sequence neural models: the Koopman Analysis of Neural Networks (KANN) method. At the core of our method lies the Koopman operator, which is a linear operator that encodes the dominant features of the network latent dynamics. In practice, we compute this operator by representing the hidden states of the network in a basis, and the operator is defined to be the linear fit in that new space. Since it is a linear operator, we can study its eigenvectors and eigenvalues, and we observe they facilitate understanding: in the sentiment analysis problem, the eigenvectors highlight positive and negative $n$-grams; and, in the ECG classification challenge, the eigenvectors capture the salient features of normal beat signals.

## 1 INTRODUCTION

Understanding the inner workings of predictive models is an essential requirement in many fields across science and engineering. This need is even more important nowadays with the emergence of neural networks whose visualization and interpretation is inherently challenging. Indeed, modern computational neural models often lack a commonly accepted knowledge regarding their governing mathematical principles. Consequently, while deep neural networks may achieve remarkable results on various complex tasks, explaining their underlying decision mechanisms remains a challenge. The goal of this paper is to help bridge this gap by proposing a new framework for the approximation, reasoning, and understanding of sequence neural models.

Sequence models are designed to handle time series data originating from images, text, audio, and other sources of information. One approach to analyzing sequence neural networks is through the theory and practice of dynamical systems (Doya, 1993a; Pascanu et al., 2013). For instance, the temporal asymptotic behavior of a dynamical system can be described using the local analysis of its attractor states (Strogatz, 2018). Similarly, recurrent models have been investigated in the neighborhood of their fixed points (Sussillo & Barak, 2013), leading to work that interprets trained RNNs for tasks such as sentiment analysis (Maheswaranathan et al., 2019). However, the local nature of these methods is a limiting factor which may lead to inconsistent results. Specifically, their approach is based on fixed-point analysis which allows to study the dynamical system in the neighborhood of a fixed-point. In contrast, our approach is global—it does not depend on a set of fixed-points, and it facilitates the exploration of the dynamics near and further away from fixed points.

Over the past few years, a family of data-driven methods was developed, allowing to analyze complex dynamical systems based on Koopman (1931) theory. These methods exploit a novel observation by which nonlinear systems may be globally encoded using infinite-dimensional but *linear* Koopman operators. In practice, Koopman-based approaches are lossy as they compute a finite-dimensional approximation of the full operator. Nevertheless, it has been shown in the fluid dynamics (Azencot et al., 2020; Mezić, 2005) and geometry processing (Sharma & Ovsjanikov, 2020; Ovsjanikov et al., 2012) communities that the dominant features of general nonlinear dynamical systems can be captured via a single matrix per system, allowing e.g., to align time series data (Rahamim & Talmon, 2021). Thus, we pose the following research question: can we design and employ a Koopman-based approach to analyze and develop a fundamental understanding of deep neural models?

Given a trained sequence neural network and a procedure to extract its hidden states, our Koopman-based method generates a moderate size matrix which faithfully describes the dynamics in the latent space. Unlike existing work, our approach is global and independent of a particular latent sample, and thus it can be virtually applied to any hidden state. A key advantage of our framework is that we can directly employ linear analysis tools on the approximate Koopman operator to reason about the associated neural network. In particular, we show that the eigenvectors and eigenvalues of the Koopman matrix are instrumental for understanding the decision mechanisms of the model. For instance, we show in our results that the dominant eigenvectors carry crucial *semantic knowledge* related to the problem at hand. Moreover, the eigenvalues represent the *memory* of the network as they provide a timestamp for the temporal span of the respective eigenvectors. Finally and most importantly, Koopman-based methods such as ours are backed by rich theory and practice, allowing us to exploit the recent advances in Koopman inspired techniques for the purpose of developing a comprehensive understanding of sequence neural networks. Thus, the **key contribution** in this work is the novel application of Koopman-based methods for understanding sequential models, and the extraction of high-level interpretable and insightful understandings on the trained networks.

We focus our investigation on two learning tasks: sentiment analysis and electrocardiogram (ECG) classification. We will identify four eigenvectors in the sentiment analysis model whose roles are to highlight: positive words, negative words, positive pairs (e.g., "not bad"), and negative pairs. In addition, we demonstrate that the eigenvectors in the ECG classification task naturally identify dominant features in normal beat signals and encode them. Specifically, we show that four Koopman eigenvectors accurately capture the local extrema points of normal beat signals. These extrema points are fundamental in deciding whether a signal is normal or anomalous. Our results reinforce that the network indeed learns a robust representation of normal beat signals. Then, we will verify that the main components of the nonlinear network dynamics can be described using our Koopman matrices by measuring the difference in accuracy results, and the relative error in predicted states. Further, we provide additional results and comparisons in the supplementary material. Given the versatility of our framework and its ease of use, we advocate its utility in the analysis and understanding of neural networks, and we believe it may also affect the design and training of deep models in the future.

## 2 RELATED WORK

**Recurrent Neural Networks (RNN) and Dynamical Systems (DS).** Fully connected recurrent neural networks are universal approximators of arbitrary dynamical systems (Doya, 1993b). Unfortunately, RNNs are well-known to be difficult to train (Bengio et al., 1993; Pascanu et al., 2013), and several methods adopt a DS perspective to improve training via gradient clipping (Pascanu et al., 2013), and constraining the weights (Erichson et al., 2021), among other approaches. Overall, it is clear that dynamical systems are fundamental in investigating and developing recurrent networks.

**Understanding RNN.** Establishing a deeper understanding of recurrent networks is a long standing challenge in machine learning. To this end, Karpathy et al. (2015) follow the outputs of the model to identify units which track brackets, line lengths, and quotes. Recently, Chefer et al. (2020) proposed an approach for computing relevance scores of transformer networks. Perhaps mostly related to our approach is the analysis of recurrent models around their fixed points (Sussillo & Barak, 2013). This approach revealed low-dimensional attractors in the sentiment analysis task (Maheswaranathan et al., 2019), which allowed to deduce simple explanations of the decision mechanisms of the associated models. Our work generalizes the approach of Sussillo & Barak (2013) in that it yields global results about the dynamics, and it introduces several novel features. We provide a more detailed comparison between our method and theirs in Sec. 4.

**Koopman-based Neural Networks.** Recently, several techniques that combine neural networks and Koopman theory were proposed, mostly in the context of *predicting* nonlinear dynamics. For example, Takeishi et al. (2017); Morton et al. (2018) optimize the residual sum of squares of the predictions the operator makes, Lusch et al. (2018); Erichson et al. (2019); Azencot et al. (2020) design dynamic autoencoders whose central component is linear and may be structured, Li et al. (2020) employ graph networks, and Mardt et al. (2018) use a variational approach on Markov processes. A recent line of work aims at exploiting tools from Koopman theory to analyze and improve the training process of neural networks (Dietrich et al., 2020; Dogra & Redman, 2020;

Manojlović et al., 2020). To the best of our knowledge, our work is first to employ a Koopman-based method towards the analysis and understanding of trained neural networks.

## 3 METHOD

In what follows, we present our method for analyzing and understanding sequence neural models. Importantly, while we mostly discuss and experiment with recurrent neural networks, our approach is quite general and applicable to any model whose inner representation is a time series. We consider neural models that take input instances $x_t \in \mathbb{R}^m$ at time $t$ and compute

$$h_t = F(h_{t-1}, x_t), \quad t = 1, 2, \dots, \tag{1}$$

where $h_t \in \mathbb{R}^k$ is a (hidden) state that represents the latent dynamics, and $F$ is some nonlinear function that pushes states through time. In our analysis, we use only the hidden states set and discard the time series input. Thus, our method is a "white-box" approach as we assume access to $\{h_t\}$, which is typically possible in most day-to-day scenarios. Importantly, all recurrent models including vanilla RNN (Elman, 1990), LSTM (Hochreiter & Schmidhuber, 1997), and GRU (Cho et al., 2014), as well as Attention Models (Bahdanau et al., 2015; Vaswani et al., 2017), and Residual neural networks (He et al., 2016) exhibit the structure of Eq. (1).

### 3.1 ESSENTIALS OF KOOPMAN THEORY

Our approach is based on Koopman (1931) theory which was developed for dynamical systems. The key observation of Koopman was that a finite-dimensional nonlinear dynamics can be fully represented using an infinite-dimensional but *linear* operator. While the theoretical background is essential for developing a deep understanding of Koopman-based approaches, the practical aspects are more important to this work. Thus, we briefly recall the definition of the *Koopman operator*, and we refer the reader to other, comprehensive works on the subject (Singh & Manhas, 1993; Eisner et al., 2015). Formally, we assume a discrete-time dynamical system $\varphi$ acting on a compact, inner-product space $\mathcal{M} \subset \mathbb{R}^m$,

$$z_{t+1} = \varphi(z_t), \quad z_t \in \mathcal{M}, \quad t = 1, 2, \dots, \tag{2}$$

where $t$ is an integer index representing discrete time. The dynamics $\varphi$ induces a linear operator $\mathcal{K}_\varphi$ which we call the Koopman operator, and it is given by

$$\mathcal{K}_\varphi f(z_t) := f(z_{t+1}) = f \circ \varphi(z_t), \tag{3}$$

where $f : \mathcal{M} \to \mathbb{R}$ is a scalar function in a bounded inner product space $\mathcal{F}$. It is easy to show that $\mathcal{K}_\varphi$ is linear due to the linearity of composition, i.e., given $\alpha, \beta \in \mathbb{R}$ and $f, g \in \mathcal{F}$, we obtain that $\mathcal{K}_\varphi(\alpha f + \beta g) = (\alpha f + \beta g) \circ \varphi = \alpha f \circ \varphi + \beta g \circ \varphi = \alpha \mathcal{K}_\varphi(f) + \beta \mathcal{K}_\varphi(g)$. We emphasize that while $\varphi$ describes the system evolution, $\mathcal{K}_\varphi$ is a transformation on the space of *functions*. From a practical viewpoint, these functions may be interpreted as observations of the system, such as velocity, sea level, temperature, or hidden states in our setup.

To justify our use of Koopman theory and practice in the context of neural networks, we propose the following. We interpret the input sequence $\{x_t\}$ as governed by some complex and unknown dynamics $\varphi$, i.e., $x_{t+1} = \varphi(x_t)$ for every $t$. We emphasize that $\varphi$ is different from $F$ in Eq. (1) by its definition of domain and range. Then, the hidden states $h_t$ are finite samplings of observations of the system, namely, $h_t \approx f_t$ where $f_t : \mathcal{M} \to \mathbb{R}$ is the true observation. For instance, $f_t$ may be the smooth function $\cos(tz)$, whereas $h_t \in \mathbb{R}^k$ is its sampling at a finite set of points $\{z_1, \dots, z_k\}$. It follows that $\{h_t\}$ is subject to an approximate Koopman representation. However, a fundamental challenge in facilitating Koopman theory in practice is the infinite-dimensionality of $\mathcal{K}_\varphi$. Recently, several data-driven methods were developed to produce a better approximate $\mathcal{K}_\varphi$ using a moderate-size matrix $C$ (Schmid, 2010; Ovsjanikov et al., 2012). In particular, Koopman-based approaches have been proven instrumental in the analysis of fluid dynamics data (Brunton et al., 2021), and for computing complex nonrigid isometric maps between shapes (Ovsjanikov et al., 2016). Motivated by these empirical examples and their success, we will compute in this work approximate Koopman operator matrices $C$ such that they encode the evolution of latent states $\{h_t\}$.

### 3.2 A KOOPMAN-BASED METHOD

We denote by $H \in \mathbb{R}^{s \times n \times k}$ a tensor of hidden state sequences, where $s$ is the batch size, $n$ is the sequence length and $k$ is the hidden dimension. The method we employ for computing the matrix $C$ follows two simple steps: 1. Represent the states using a basis $B$, and denote the resulting collection of spectral coefficients by $\tilde{H}$. 2. Find the best linear transformation $C$ which maps $\tilde{H}_t$ to $\tilde{H}_{t+1}$ in the spectral domain, where $\tilde{H}_\tau \in \mathbb{R}^{s \times k}$ denotes the tensor of coefficients from $\tilde{H}$ at time $\tau$. To give a specific example of the general procedure we just described, we can choose the principal components $b_j, j = 1, 2, ...$ of the truncated SVD of the states $H$ to be the basis in the first step. Then, the resulting basis elements are orthonormal, i.e., $B^T B = \text{Id}$, where $B = (b_j)$ is the matrix of basis elements organized in its columns, and $\text{Id}$ is the identity matrix. The matrix $C$ is obtained by solving the following least squares minimization

$$C := \arg\min_{\tilde{C}} \sum_{t=1}^{n-1} \left| \tilde{H}_t \cdot \tilde{C} - \tilde{H}_{t+1} \right|_F^2 , \tag{4}$$

$$\tilde{H}_\tau = H_\tau \cdot B , \quad \forall \tau , \tag{5}$$

where $\cdot$ is matrix multiplication. We note that the above scheme is a variant of the dynamic mode decomposition (Schmid, 2010), and the functional maps (Ovsjanikov et al., 2012) algorithms.

### 3.3 KOOPMAN-BASED PREDICTION

The infinite-dimensional Koopman operator in Eq. (3) describes the evolution of observable functions subject to the dynamics $\varphi$. Similarly, our $C$ matrices allow us to predict a future hidden state $h_{t+1}$ from a given current state $h_t$ simply by multiplying $C$ with the spectral coefficients $\tilde{h}_t$. Namely,

$$H_{t+1}^{\text{KANN}} := H_t \cdot B \cdot C \cdot B^T . \tag{6}$$

We will mostly use Eq. (6) to evaluate the validity of $C$ in encoding the underlying dynamics based on the differences $|H_t^{\text{KANN}} - H_t|_F^2 / |H_t|_F^2$ for every admissible $t$, see Sec. 4.

### 3.4 KOOPMAN-BASED ANALYSIS

The key advantage of Koopman theory and practice is that linear analysis tools can be directly applied to study the behavior of the underlying dynamical system. The tools we describe next form the backbone of our analysis framework, and our results are heavily based on these tools.

**Separable dynamics.** If $C \in \mathbb{R}^{k \times k}$ admits an eigendecomposition, then the dynamics can be represented in a fully *separable* manner, where the eigenvectors of $C$ propagate along the dynamics independently of the other eigenvectors, scaled by their respective eigenvalues. Formally, we consider the eigenvalues $\lambda_j \in \mathbb{C}$ and eigenvectors $v_j \in \mathbb{C}^k$ of $C$, i.e., it holds that $C v_j = \lambda_j v_j$. We assume that $C$ is full-rank and thus $V = (v_j)$ forms a basis of $\mathbb{R}^k$, and similarly, $U = V^{-1}$ is also a spanning basis. In our setting, we call the rows of $U$ the **Koopman eigenvectors**, and we represent any hidden state $h_t$ in this basis, similarly to Eq. (5). The projection of $H$ onto $U$ reads

$$\hat{H}_\tau := H_\tau \cdot B \cdot V = \tilde{H}_\tau \cdot V . \tag{7}$$

Then, re-writing Eq. (6) using the eigendecomposition of $C = V \cdot \Lambda \cdot U$ yields the temporal trajectory of $H_t$ via $\hat{H}_{t+1} = H_{t+1} \cdot B \cdot V \approx H_t \cdot B \cdot C \cdot V = H_t \cdot B \cdot V \cdot \Lambda = \hat{H}_t \cdot \Lambda$, where $\Lambda$ is the diagonal matrix of eigenvalues, and the approximation is due to Eq. (6). The latter derivation yields

$$\hat{H}_{t+1} \approx \hat{H}_t \cdot \Lambda , \tag{8}$$

i.e., the linear dynamics matrix represented in the basis $U$ is simply the diagonal matrix $\Lambda$, and thus $U$ may be viewed as a "natural" basis for encoding the dynamics. Further, it directly follows that $\hat{H}_{t+l} \approx \hat{H}_t \cdot \Lambda^l$, that is, the number of steps forward is determined by the eigenvalues power.

**Memory horizon.**   In addition to their role in describing the evolution of the hidden states, the eigenvalues naturally encode a memory horizon "timestamp" associated with each of the eigenvectors. Specifically, if $|\lambda_j| < 1$, then its powers decay exponentially to zero. We define the memory horizon of $u_j$ to be the decay time $\tau_j(\epsilon)$ of its associated eigenvalue, i.e.,

$$\tau_j(\epsilon) = \frac{\log(\epsilon)}{\log(|\lambda_j|)} \ , \tag{9}$$

where $0 < \epsilon \ll 1$ is a small threshold parameter. We note that both $\log(\epsilon)$ and $\log(|\lambda_j|)$ are negative, and thus $\tau_j(\epsilon) > 0$. Given a choice of $\epsilon$, the eigenvector $u_j$ becomes almost insignificant to the dynamics in Eq. (9) after $\tau_j(\epsilon)$ steps as it is scaled by $\epsilon = |\lambda_j|^{\tau_j(\epsilon)}$. In this context, eigenvectors whose eigenvalues satisfy $|\lambda_j| = 1$ have *infinite* memory, as powers of their eigenvalues simply rotate over the unit circle. Finally, if $|\lambda_j| > 1$, its respective eigenvector leads to an unstable behavior since $\lim_{l \to \infty} |\lambda_j^l| = \infty$.

## 4   RESULTS

In this study, we focus our exploration on the sentiment analysis and the ECG classification problems. Unless noted otherwise, we always compute $C$ using the method in Sec 3.2, where the basis is given by the truncated SVD modes of the input hidden states, and $C$ is the least squares estimation obtained from solving (4). We first provide our qualitative analysis in 4.1, and 4.2. Then, we include in 4.3 a quantitative evaluation of KANN and its ability to encode the dynamics. In Apps. A, D, and E, we provide additional results, and we show that our method is robust to the choice of basis and network architecture. Finally, we further use KANN to analyze the copy problem in App. F, where our results outperform the baseline approach (Maheswaranathan et al., 2019).

### 4.1   SENTIMENT ANALYSIS

We begin our qualitative study by considering the sentiment analysis task which was extensively explored in (Maheswaranathan et al., 2019; Maheswaranathan & Sussillo, 2020). Determining the sentiment of a document is an important problem which may be viewed as a binary classification task. We will use the IMDB reviews dataset, and we will embed the corpus of words to obtain a vector representation of text. Given a review, the role of the network is to output whether it reflects a positive or negative opinion. Adopting the setup of Maheswaranathan et al. (2019), we use a word embedding of size 128, and a GRU recurrent layer with a hidden size of 256. We train the model for 5 epochs during which it reaches an accuracy of $\approx 92\%, 87\%, 87\%$ on the train, validation and test sets, respectively. For analysis, we extract a random test batch of 64 reviews and its states $H \in \mathbb{R}^{64 \times 1000 \times 256}$, where 1000 is the review length when padded with zeros.

One of the main results in (Maheswaranathan et al., 2019) was the observation that the dynamics of the network span a line attractor. That is, the hidden states of the network are dominantly attracted to a one dimensional manifold, splitting the domain into positive and negative sentiments. Additionally, Maheswaranathan & Sussillo (2020) study inputs with contextual relations (e.g., the phrase "not bad"), and their effect on the network dynamics. Our results align with the observations in (Maheswaranathan et al., 2019; Maheswaranathan & Sussillo, 2020). Moreover, we generalize their results by showing that the attracting manifold is in fact of a higher dimension, and that the manifold can be decomposed to semantically understandable components using KANN. Specifically, we demonstrate that several Koopman eigenvectors are important in the dynamics, and we can link each of these eigenvectors to a semantically meaningful action. Thus, in comparison to the framework proposed in (Maheswaranathan et al., 2019; Maheswaranathan & Sussillo, 2020), our method naturally decomposes the latent manifold into interpretable attracting components. In addition, we provide a unified framework for reasoning and understanding by drawing conclusions directly from the separable building blocks of the latent dynamics.

Most of our results for the sentiment analysis problem are based on the eigendecomposition of $C$, its resulting eigenvalues $\{\lambda_j \in \mathbb{C}\}$ and corresponding eigenvectors $\{u_j \in \mathbb{C}^k\}$. For the random states batch $H$ specified above, we obtain an operator $C$ whose spectrum consists of *four* eigenvalues with modulus greater than 0.99, i.e., $|\lambda_j| > 0.99$. In comparison, Maheswaranathan et al. (2019) identify only a single dominant component. The values of our $\lambda_j$ read $\lambda_1 = 0.9999$. $\lambda_2 = 0.9965$, and

A positive review projected onto $\{u_1, u_2\}$:

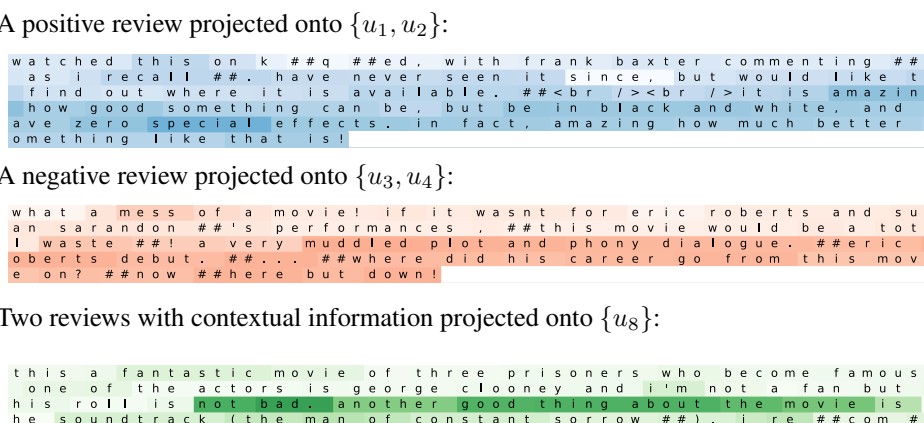

A negative review projected onto $\{u_3, u_4\}$:

Two reviews with contextual information projected onto $\{u_8\}$:

Figure 1: We display reviews where each word is shaded based on $\sum_j s(j, h_t)$. Projecting onto $U_{12}$ shows an increase in magnitude for several positive words (blue), whereas projecting onto $U_{34}$ shows jumps in magnitude around negative words (red). See e.g., `amazing`, `special` (blue), `mess`, `waste`, `muddled` (red). We also show two reviews with contextual information which is naturally highlighted due to $u_8$.

$\lambda_{3,4} = 0.9942 \pm i0.0035$. Consequently, their respective eigenvectors have long memory horizons. For instance, if we set $\epsilon = 1e{-}2$ in Eq. (9), then we get a time stamp $\tau > 800$ for all four eigenvectors. Namely, these eigenvectors carry information across word sequences of length up to $800$, and only $< 2\%$ of the reviews in the IMDB dataset are that long.

In their analysis, the authors of (Maheswaranathan et al., 2019) observe that the network mainly counts positive vs. negative words in a review along a line attractor. We hypothesize that in our setting, the dominant eigenvectors $\{u_1, ..., u_4\}$ are responsible for this action. We expect that complex conjugates such as $u_3$ and $u_4$ share the same role, e.g., counting the negative words, and we expect that $U_{12} = \{u_1, u_2\}$ take the role of counting the positive words. To verify our hypothesis, we use the readout layer of the model to generate the logits of the state when projected to $U_{12}$ and $U_{34}$. We denote by $\tilde{y}_{12}$ and $\tilde{y}_{34}$ the logits for $\tilde{H} \cdot V_{12}$ and $\tilde{H} \cdot V_{34}$, respectively, where $V_{ij}$ denote the $i$ and $j$ columns of $V$. For the above test batch, we get perfect correspondence, i.e., $\tilde{y}_{12} < .5$ and $\tilde{y}_{34} > .5$ on all samples. In addition to encoding a certain sentiment, the Koopman eigenvectors are advantageous in comparison to a single line attractor as they allow for a direct visualization of the importance of words in a review. Specifically, we define the projection magnitude of a hidden state as follows

$$s(j, h_t) := \left| \hat{h}_t(j) \right| = \left| \tilde{h}_t^T V_j \right| . \tag{10}$$

We show in Fig. 1 two examples where the magnitude of projection onto $U_{12}$ and $U_{34}$ clearly highlights positive and negative words, respectively. In particular, as the network "reads" the review and identifies e.g., a negative word, it increases $s(\cdot)$. For instance, see `mess` and `muddled` in Fig. 1. Importantly, there may be occurrences of positive/negative $n$-grams which are not highlighted, such as the word `good` in the positive example. We show in App. A that the above results extend to the entire test set. Thus, we conclude that $\{u_1, ..., u_4\}$ track positive and negative words.

In addition to $\{u_1, ..., u_4\}$, we also want to understand how other eigenvectors affect the latent dynamics. We hypothesize that other vectors are responsible to track contextual information such as amplifiers ("extremely good") and negations ("not bad"). We collected all reviews that include the phrases "not bad" and "not good" into a single batch, yielding a states tensor with 256 samples. One way to check our hypothesis is to employ the former visualization using other eigenvectors. We show two such examples in Fig. 1 in green, where phrases such as "not bad", "terribly wrong", and

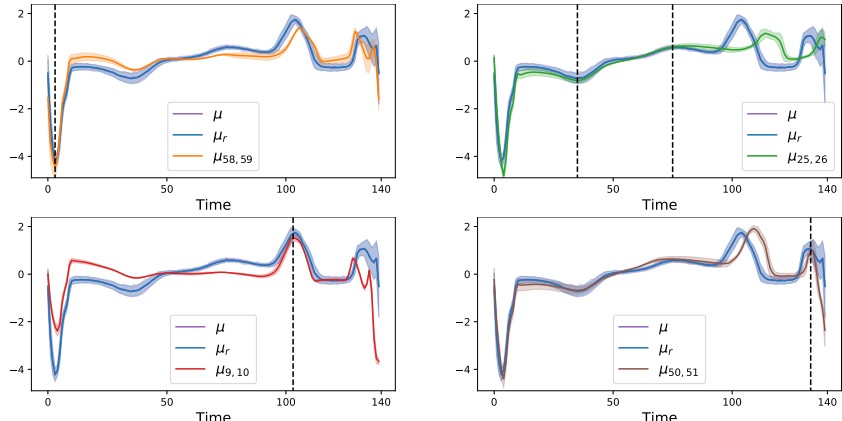

Figure 2: We show the median of reconstructions of normal beats when projected to each of the first four dominant conjugate pairs of Koopman eigenvectors. The medians (orange, green, red, brown) are plotted on top of the original signals and their reconstruction medians. The dashed black lines indicate important features of the signals which are well captured by the Koopman eigenvectors.

"quite ok" are highlighted when projected onto $u_8$. We provide additional results and analysis on the identification of amplifier words and negations using KANN in App. A. In addition to unigram and bigram highlighting, in App. B we also consider KANN in the general case of $n$-grams where $n > 2$.

## 4.2 ECG CLASSIFICATION

Electrocardiogram (ECG) tests track the electrical activity in the heart, and they help detect various abnormalities in a non-invasive way. Classifying whether a beat is normal or not is a challenging task which lacks descriptive neural models. A common approach for solving the classification problem using neural networks trains an autoencoder model with an $L^1$ loss over the normal beats. Classification is performed by measuring the loss between the original and reconstructed signals; thus, while it is a classification task, ECG classification is solved via a *regression* model. In particular, high loss values indicate anomalous beats, whereas low values are attributed to normal signals. Typically, a threshold is set during the training phase, allowing automatic classification on the test set. We fix the threshold to be 26. Our network is composed of a single layer LSTM encoder $F_{\text{enc}}$ with a hidden size of 64, and an LSTM decoder $F_{\text{dec}}$ with one layer as well. We use a publicly available subset of the MIT-BIH arrhythmia database (Goldberger et al., 2000) for our data, named ECG5000[1]. This dataset includes 5000 sample heartbeats with a sequence length of 140. Around 60% of the sequences are classified as normal and the rest are various anomalous signals. The model is trained for 150 epochs, yielding an accuracy of 97.1%, 97.6%, 98.6% on the train, validation and test sets, respectively.

Similarly to the sentiment analysis problem, we expect the Koopman eigenvectors to take a significant role in encoding the latent dynamics. Given that the network is generative as it is an autoencoder, we hypothesize that the eigenvectors $\{u_j\}$ capture dominant features of normal beats. Thus, we project normal beats onto pairs of dominant eigenvectors, and decode the resulting hidden states using the decoder to study the obtained signals. For example, say $U_{58,59}$ are dominant, then we project onto the space spanned by this pair, then project back and decode. Using a test batch of 64 *normal* beats we collect the last hidden state of every sample in a matrix $H_{\text{no}} \in \mathbb{R}^{64 \times 64}$ and we compute the following.

$$\mathbb{R}^{64 \times 140} \ni \bar{X}^{ij} = F_{\text{dec}}(H_{\text{no}} \cdot B \cdot |V_{ij} \cdot U_{ij}| \cdot B^T), \tag{11}$$

where $|\cdot|$ is the element-wise modulus of complex numbers. To determine which eigenvectors are dominant, we employ Eq. (10).

To visualize the results, we take the set of reconstructed signals $\bar{X}^{ij}$ of a particular pair $ij$, and we compute its median $\mu_{ij}(t)$ for $t \in [1, \ldots, n]$. We plot these graphs in Fig. 2 using colors orange (pair 58–59), green (pair 25–26), red (pair 9–10), and brown (pair 50–51). In addition, the original signals'

---

[1]http://timeseriesclassification.com/description.php?Dataset=ECG5000

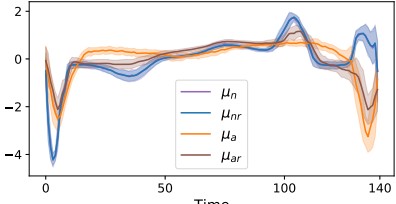 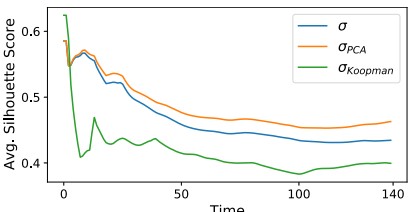

Figure 3: The network reconstructs relatively well both normal and anomalous signals (left), implying its inner representation is binary. However, the silhouette scores of the Koopman embedding (right) imply that only a single class is being learnt in practice.

median $\mu$ (purple) and the median of the signals reconstruction $\mu_r$ (blue) are shown in each of the subplots for comparison. Indeed, $\mu$ and $\mu_r$ are almost indistinguishable, and can be differentiated only when zooming in. Each median graph is wrapped in its median absolute deviation envelope. We preferred median-based quantities over the common mean and standard deviation since the latter produce cluttered plots in our setting due to outliers. The plots in Fig. 2 clearly show that each conjugate pair captures a different feature of the time series as marked by the vertical dashed lines. Specifically, $\mu_{58,59}$ captures the minimum around $t = 3$, and $\mu_{25,26}$ encodes the part of the signal in $t \in [35, 75]$. Moreover, $\mu_{9,10}$ attains the maximum at $t = 103$, and $\mu_{50,51}$ is approximating the lower peak at $t = 133$ and we consider these $t$ values to be the salient features. Importantly, the other Koopman eigenvectors beyond the ones we consider above are less important in the reconstruction, and are mostly helpful in fixing minor variations. Finally, we provide a similar computation in App. D using the dominant `PCA` modes and `KernelPCA` eigenvectors, where we show that `PCA` components and `KerenelPCA` eigenvectors are not useful in identifying the dominant features of beat signals. Also, we provide a quantitative comparison between the methods.

In addition to identifying principal features of beat signals, we show in what follows that the Koopman eigenvectors are also instrumental in analyzing the latent structure of the LSTM autoencoder. We begin by showing in Fig. 3 (left) the median values over time of a normal batch and its reconstruction (as in Fig. 2), and similarly for a batch of anomalous signals (orange and brown). From this data, the task of ECG classification may be viewed as a binary classification problem, separating normal from anomalous signals via reconstruction. However, we will now show that this is actually not the case. Instead, the network essentially encodes inputs, whether normal or anomalous, in a representation that is closer to the manifold of *normal* signals. To demonstrate and analyze this phenomenon, we consider hidden state tensors $H_{no}$ and $H_{an}$ of normal and anomalous beats, respectively, and we concatenate these tensors over the samples yielding $H = (H_{no}, H_{an}) \in \mathbb{R}^{128 \times 140 \times 64}$. We would like to study the decision boundary separating between different signals in the latent space.

To this end, we employ a standard measure known as the silhouette score (Rousseeuw, 1987) to quantify the class separation quality. The silhouette score $\sigma$ is a real value in $[-1, 1]$, where scores close to 1 mean the latent states are well separated. In contrast, values closer to zero indicate that samples are located on or close to the decision boundary. We compute the silhouette score estimates on $\tilde{H} = H \cdot B$ averaged over samples, and cumulatively averaged over time. Namely, $\sigma(t) = \frac{1}{128 \cdot t} \Sigma_{s,t} \sigma(\tilde{h}_{s,t})$, where $\sigma(\tilde{h}_{s,t})$ is the silhouette score of the vector $\tilde{h}_{s,t}$ with $s \in [1, \ldots, 128]$ and $t \in [1, \ldots, 140]$. We compare three silhouette score estimates denoted by: $\sigma$ for the original $\tilde{H}$, $\sigma_{\text{PCA}}$ for the projection of $\tilde{H}$ onto its first five principal components, and $\sigma_{\text{Koopman}}$ for the projection of $\tilde{H}$ onto the first five dominant Koopman eigenvectors. The results are shown in Fig. 3 (right), where Koopman's embedding attains low scores compared to `PCA` and the original states. Namely, embedding the hidden states using Koopman eigenvectors reveals that the decision boundary between normal and anomalous signals is somewhat blurred, in contrast to the numerical results provided by the reconstructed signals and other embeddings. This understanding provides a rather straightforward interpretation of the model: it simply encodes the dominant components of all signals as being normal, allowing to easily identify anomalous signals later by measuring their reconstruction error. Finally, our analysis shows that Koopman eigenvectors successfully identify the salient features of normal beat signals. We conclude from this observation that the network focuses on identifying these features and reconstructing them accurately. A correct reconstruction of these salient features allows to subsequently identify using a simple loss check whether a signal is normal or anomalous. Importantly,

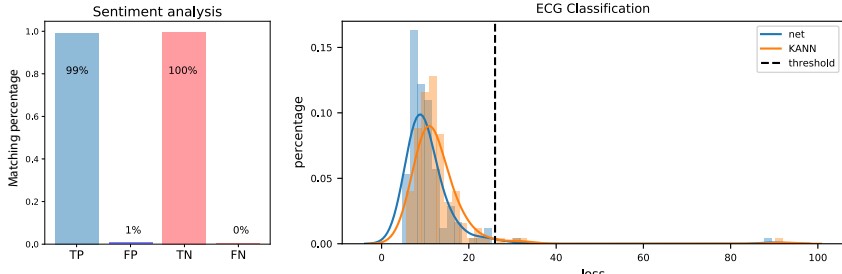

Figure 4: On the tasks we consider, our approach approximately reproduces the network classification outputs. In particular, we obtain $> 99\%$ and $> 97\%$ agreement on the sentiment analysis and ECG classification problems, respectively. See the text for additional details.

we show in Fig. 3 that the network indeed successfully reconstructs normal and anomalous signals. The understanding that we obtain using KANN is that the network mainly focuses on these salient features during the reconstruction.

### 4.3 KANN REPRODUCES THE LATENT DYNAMICS

We now perform a quantitative study of the ability of $C$ to truly capture the latent dynamics. We will show that indeed, KANN is able to reproduce the nonlinear dynamics of the network in Eq. (1) to a high degree of precision, and thus we achieve the empirical justification to replace $F$ with $C$. To this end, we consider the following two metrics:

1. **Relative error** of hidden states: let $\{h_{s,t}\}$ be a collection of states over samples $s = 1, ..., S$ and across time $t = 1, ..., T$. We generate the predicted collection $\{h_{s,t}^{\text{KANN}}\}$ using Eq. (6), and we compute

$$e_{\text{rel}}(\{h_{s,t}^{\text{KANN}}\}, \{h_{s,t}\}) = \frac{1}{T \cdot S} \sum_{s,t} |h_{s,t}^{\text{KANN}} - h_{s,t}|_2^2 \,/\, |h_{s,t}|_2^2 \,. \tag{12}$$

2. **Accuracy error**: let $G$ be the neural network component that takes a state and produces the output of the model, i.e., $G(h_t) = \tilde{y}_t$. We denote by $\tilde{c}_t$ the category predicted by $\tilde{y}_t$, for instance $\tilde{c}_t = \arg\max(\tilde{y}_t)$. We compare the difference between $\tilde{c}_t$ and $\tilde{c}_t^{\text{KANN}}$, obtained from $G(h_t^{\text{KANN}}) = \tilde{y}_t^{\text{KANN}}$.

We show in Fig. 4 the results of our quantitative study. For the sentiment analysis problem (Fig. 4, left), we obtain $> 99\%$ correspondence with the classification of the network over *all* the test set, as is shown for the True Positive (TP) and True Negative (TN) columns vs. the False Positive (FP) and False Negative (FN) columns. In the ECG classification task (Fig. 4, right), we reconstruct 145 signals of the normal test set and compute their loss. There is a noticeable yet small shift in the loss histogram between the network reconstruction (blue) in comparison to our reconstruction (orange). However, the threshold for this problem set at 26 during training (black dashed line) yields $> 97\%$ agreement in classification. In particular, the false classification of normal signals (around loss 90) appear both in the network output and in ours. Finally, we also computed the relative error of the hidden states, obtaining $e_{\text{rel}} = 0.095$ on a batch of size 64 for the sentiment analysis task, and $e_{\text{rel}} = 0.0056$ on a batch of size 145 for the ECG classification problem. Overall, the above results demonstrate that KANN faithfully represents the latent dynamics.

## 5 DISCUSSION

In this work we presented a novel framework for studying sequence neural models based on Koopman theory and practice. Our method involves a dimensionality reduction representation of the states, and the computation of a linear map between the current state and the next state. Key to our approach is the wealth of tools we can exploit from linear analysis and Koopman-related work. In particular, we compute linear approximations of the state paths via simple matrix-vector multiplications. Moreover, we identify dominant features of the dynamical system and their effect on inference and prediction. Our results on the sentiment analysis problem, and the ECG classification challenge provide simple yet accurate descriptions of the underlying dynamics and behavior of the recurrent models. Our work lays the foundations to further develop application-based descriptive frameworks, towards an improved understanding of neural networks. In the future, we plan to explore our framework during the training of the model, where in this work we focused only on trained models.

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
