# OpenReview forum: "A Koopman Approach to Understanding Sequence Neural Models"
_ICLR.cc/2022/Conference — ICLR 2022 Submitted_

### Official Review · Reviewer_Dka9 · 2021-11-01

**Correctness:** 4
**Technical Novelty And Significance:** 1
**Empirical Novelty And Significance:** 2
**Recommendation:** 3
**Confidence:** 4

**Main Review:**

Strengths:
1. Applies the Koopman operator framework to develop some insights into two sequence modeling problems.

Comments:

1. My main issue with this paper is that this method is not new, and it doesn't really add anything from a methods standpoint. It's more that it applies an existing framework (I would not agree this is novel) to some model/prediction tasks which it has not been before.
2. It seems like it would be a better fit for an applied conference focusing on the problems in question (e.g., sentiment analysis). However, even then, I think it would be necessary to apply the Koopman analysis to several datasets and have more developed insights using the Koopman analysis.


**Summary Of The Paper:**

The paper applies the developed theory of the Koopman operator to analyzing neural sequential models empirically, on two tasks: ECG and sentiment analysis, and derives some insights on what the models are doing using the spectrum of the Koopman operator.


**Summary Of The Review:**

Overall, I lean on the rejection side: the paper is not that novel techniques-wise in my opinion, and the insights on the applications need to be more developed and thorough.


UPDATE:

I read the other reviews and responses and my opinion is unchanged: The final response to my questions was not convincing overall in terms of the insights provided.

---

> ### Author Response · Authors · 2021-11-10
> **Response to Reviewer Dka9**
>
> We appreciate your feedback. As we clearly specify in the paper, the tool (applied Koopman operator) we use is not new. The main contributions in our paper are the application of Koopman methods in the context of understanding sequence neural models, and the analysis on sentiment analysis and ECG classification. To the best of our knowledge, this has not been done before. We would appreciate to receive feedback regarding the novel contributions of our paper, and we anticipate your response. In particular, we would like to better understand what you mean by "the applications need to be more developed and thorough".

---

> > ### Comment · Reviewer_Dka9 · 2021-11-10
> > **regarding applications**
> >
> > As you mentioned, the main novelty in this paper is the application of the Koopman framework to two tasks (sentiment analysis and ECG).
> > My issue is that in both of these cases, I did not think the resulting insights were very interesting -- also, applying the method to only two specific tasks, and then not really drawing too many conclusions from the application of the method, does not meet the bar for a conference paper in my opinion. It would be a much stronger paper if you applied the method to more problems and uncovered particular, useful structure in each of those problems that couldn't be otherwise uncovered by previous methods.  As is, my main takeaway is that Koopman methods do a decent job of fitting the dynamics of these sequence models.
> >
> > Let me restate the discoveries to make sure I understand:
> > 1) For both tasks, a small # of Koopman eigenfunctions does a decent job of approximating the dynamics,
> > 2) Previous work suggested that these sequence models are essentially counting positive/negative words, you identified some eigenvectors (subspaces) which particularly influenced counting positive words vs counting negative words and visualized this
> > 3) For ECG task, the network encodes the inputs as normal signals (I don't quite understand what you mean by this, that section of the analysis could use re-writing for clarity purposes) and determines if a signal is not normal by looking at reconstruction error (this needs to be clarified as well).
> >
> > I don't see how these findings reveal anything particularly interesting about the sequence models, nor do I see how these are useful as diagnostic tools for helping understand failure modes in neural network training (that might be a good potential idea to try). The most interesting thing here is simply that ok, you can fit the dynamics of these sequence models w/Koopman methods (even with linear basis functions), so the dynamics are in some sense simple. But I'm not sure what to take away from that.
> >
> > An example of useful takeaway might be something that gives insight into a failure of modeling, or which suggests a better way of training the model or changing the model architecture, and so on. Alternatively it would be interesting if you could clearly characterize a class of simpler algorithms for solving the problem (the closest this paper gets to that is suggesting that the previous work of Maheswaranthan et al isn't complete; there's actually a slightly higher dimensional manifold in play -- but this isn't particularly surprising or insightful -- at least, there should be more interpretation of this result in a meaningful way -- just pointing to which eigenvectors correspond to highlighting positive/negative words doesn't seem like much of an interpretation -- it's not really clear that there is a nice interpretation for this, because it's not (IMO) super meaningful).
> >
> > So these problems could be fixed if:
> > 1) there were concrete interpretations of the results beyond just saying "the Koopman method fit the dynamics, and we were able to point to slight modifications of existing results" (e.g. not just a 1 dim line attractor, but maybe 5-dim, and here are pairs of eigenvectors which correspond to the previously identified counting pos/neg mechanisms)
> > 2) there was a clearer identification of interesting structure that you can detect by using Koopman analysis perhaps across a few more problem domains (or really in depth in 1 or 2 domains; the current results are very surface level in my opinion)
> > 3) in particular it would be helpful if there were actionable insights (either in terms of understanding what might be going wrong with model training / model architecture, or in terms of understanding something nice about the overall algorithms the sequence models are using (the second approach is what you attempt to do, but I already outlined how it falls short)

---

> > > ### Author Response · Authors · 2021-11-11
> > > **regarding contribution**
> > >
> > > Thank you for your detailed and honest response. In what follows we would like to better position our paper with respect to the existing work in the sub-community of researchers who develop methods and analyze neural networks from a dynamical systems perspective. Specifically, we list some of the recent papers in this field and try to classify them. We apologize if our classification is over-simplistic, but we believe it would make it easier this way to position our paper with respect to existing work. Here is the (non-exhustive) list of papers:
> > >
> > > 1. Sussillo et al.'13: Introduced the fixed-point analysis (FFA) method. Analyzed several examples.
> > >
> > > 2. Maheswaranathan et al.'19: Used FFA. Analyzed Sentiment analysis.
> > >
> > > 3. Maheswaranathan et al.'20: Developed a Piecewise-linear method (as mentioned by Reviewer XyzC). Analyzed 2-grams in Sentiment analysis.
> > >
> > > 4. Aitken et al.'20: Used FFA. Analyzed Text classification.
> > >
> > >
> > > Given the recent developments in this sub-community and that several of them were published as conference papers, we believe that our paper advances the knowledge on this topic, and thus it makes a strong case for acceptance as a conference paper. In particular,
> > >
> > > 1. We propose a new method (KANN) to solve the problem of understanding neural networks. Our method is backed by theory and practice. The operators we compute faithfully encode the dynamics as advertised. Using a linear basis and least squares, our method is easy to compute, and it is on-par with respect to FFA in terms of computational resources.
> > >
> > > 2. We evaluate KANN on sentiment analysis, and we show it achieves similar results to FFA. More importantly, we show that using the same machinery (Koopman eigenvectors), one can attain interpretable results for 1-grams and **2-grams**. In contrast, previous approaches need to develop a tailored machinery to analyze just this feature. This makes our method easier to use with respect to existing work.
> > >
> > > 3. We evaluate KANN on ECG classification. This is a new task that was not explored before. Moreover, it is not a classifcation architecture but an autoencoder (generative) architecture, making it a regression problem, and thus differing significantly from previously analyzed tasks (sentiment analysis and text classification). Our results identify temporal structures that are not easy to identify with other means, and we observe that the network simply learns a single class, regardless of the input.
> > >
> > > 4. We provide analysis to another new task (the copy problem) in the supplmental material, as well as additional results and comparisons.
> > >
> > > From a research perspective, proposing a new method for tackling a problem is always important as it opens up new possibilities for further research and new algorithms and results. Showing the method works on previously studied cases is important as a sanity check, and the additional findings (easier interpreatation of 2-grams) is definitely a plus. Finally, exploring a new test case (ECG classification) which is different from previous tests cases (classification vs. regression), shows the method is able to extend to new regimes. To the best of our knowledge, the contributions listed above are typically sufficient to be considered for publication in most venues (ICLR included). It is unfortunate that you do not find our results exciting, but excitment aside, we believe our paper provides a detailed scientific work with method, evaluation, analysis and discussion that follow the standard scientific protocol in terms of rigor and novelty.

---

> > > > ### Comment · Reviewer_Dka9 · 2021-11-11
> > > > **reply regarding contributions**
> > > >
> > > > Thanks, I see the positioning of the paper. I suppose I buy the argument that it's perhaps a cleaner methodology to get results about interpreting the effect of 2-grams -- can you say anything about general n-grams, or more generally, understand more about what the network is doing beyond what previous work describes?
> > > >
> > > > BTW: This paper should probably also be cited: https://arxiv.org/abs/2010.15114
> > > >
> > > > Here are my remaining issues, with one overarching question: What will a reader learn from your paper other than "Koopman analysis is a reasonable method to try when analyzing network dynamics?" In particular, for the problems you analyze, what is actionable about the takeaways/understanding? And what is the result of the analysis beyond "here are some eigenfunctions which correspond to positive/negative words"? Why is it useful to know that? How can one understand something about the task, network architecture, training method, etc etc by doing this analysis? Previous work had as a novel insight that a) the linear dynamics (without global temporal scale) are usable to understand one mechanism that the network uses, b) an identification of that mechanism. This paper suggests the previous work was limited, and suggests that the dynamics are higher dimensional, but doesn't provide additional understanding using that knowledge.
> > > >
> > > > To the points:
> > > >
> > > > 1. I don't really agree that the method is new -- it's a novel application of an existing method, which yes, you point out in the paper. Without concrete interpretations of the results of the analysis, to me, this means that the role of this paper is more of an advertisement for an existing method which someone may not have heard of. I guess your mileage may vary as to whether or not this is sufficient for acceptance, so I'll leave it to the AC to decide this.
> > > > 2. Could you provide more clarification on exactly what the insights are for the ECG classification task? I really didn't understand how the Koopman analysis as presented in the paper was useful. I can kind of buy the insights for sentiment analysis, especially if they are extendable to more than just 2 n-grams (if it is just 2 n-grams, you do have a point that you recover what previous work studied in a broader framework, but I would be much more inclined to accept if for instance you were able to say something new and interesting about the impact of n-grams for larger n as well, though this seems hard for this method to surface).  But as mentioned above, I would really appreciate more explanation on why the existing results, as presented, are useful/interesting -- how would this fit into the workflow of someone trying to understand the network (either after training, or before, or whatever use case you had in mind for someone using the Koopman analysis)? (I recognize that perhaps the earlier work only partially tackled this question, which I view as a weakness of the earlier work -- but they did put forward a novel insight about the way RNNs work for sentiment analysis, whereas you merely confirm that existing analysis with a modification -- if you could interpret the results about the higher dimensional manifold more and synthesize them into an insight about how the network works beyond recovering the same insight as prior work, that would also make me much more inclined to accept).
> > > > 3. Regarding the point about research -- of course what you say is true, but what you are doing here is more akin to taking a well-established method (that has also been applied in the ML literature, albeit not for the specific problem you're dealing with) and advertising it to the community -- in such a case, it is not sufficient in my opinion to simply show that that method recovers previous results, with a few extra results that aren't really synthesized into an explanation, unless the method has the potential to create new insights. For instance, the prior work focused on highlighting the novel takeaways, rather than their application of linearizing the dynamics. If you could more strongly highlight what this method uncovers about the way these sequence models work that goes beyond existing work, that would be a strong point in this paper's favor. As it is currently presented, I don't see a significant leap in novelty over existing results.
> > > >
> > > > I wanted to also clarify that I think finding ways to use Koopman analysis for understanding neural nets is very interesting overall, it's just that I think it needs to be applied to greater effect to really determine some actionable insights in the behavior of various networks -- the current results still lack this property. If you can highlight how one might use the Koopman analysis to more deeply understand properties of the task (likely this requires doing more analysis, unless I missed something -- please point out if I did), I would be much more inclined to accept. I don't have a problem with the method not being novel as long as there are novel, actionable insights.

---

### Official Review · Reviewer_Ts2w · 2021-11-01

**Correctness:** 4
**Technical Novelty And Significance:** 3
**Empirical Novelty And Significance:** 2
**Recommendation:** 6
**Confidence:** 3

**Main Review:**

The authors propose a new scheme for understanding the representations captured by sequence neural models. They build upon the Koopman operator and provide a framework for studying dynamical NNs. The paper is well written, and the method is clearly explained. The results demonstrate that the Koopman operator can capture the essential information behind the dynamics captured by supervised sequence-based NNs.
In what follows, I will detail my comments that could help improve the manuscript.
-The abstract is not very clear, specifically “local theories” are not defined at this point, so this statement is confusing.
-On the same point, the abstract does not provide intuition on the method; please expand.
-” However, the local nature”- this statement is again not clear.
-Last paragraph of the intro: “dominant features in normal beat signal”- this is not clear at this point what you mean. I had to read the experimental part to understand. Would you please make the description in the introduction self-explanatory?
-” which is typically possible in most day-to-day” -can you add examples here?
-P3 “moderate size matric C” - the matrix C is not yet defined at this point.
-4.1 what are the sizes of the train/test/valid, also? How are the parameters of the network tuned?
-4.2 same questions as above.
-4.2 how was the threshold 26 selected?
 -”the network is generative” in what sense is it generative?
-the comparison to PCA does not completely convince me; could you quantify the advantage over PCA? Also, can you try to use the eigenvectors of Kernel-PCA?
-References from the main text to some of the appendix parts are missing, for example, Appendix C.


**Summary Of The Paper:**

The authors propose a method for analyzing NN applied to temporal signals (e.g., dynamical systems). The proposed method relies on the Koopman operator, which is widely used for studying dynamical systems. The authors use the eigenvectors of the Koopman operator to interpret predictions made by NN. The proposed method is evaluated on two tasks, namely: sentiment analysis and electrocardiogram (ECG) classification. The authors demonstrate that the eigenvectors of the Koopman operator do encode the latent information in these examples.


**Summary Of The Review:**

To summarize, the proposed method is interesting and could lead to several extensions. The English level is OK, and the writing is mostly clear. However, I believe that some of the descriptions (mainly in the results section) should be strengthened before the paper can be published. Moreover, I would expect a comparison to more baselines in the main text, like PCA or Kernel PCA. For these reasons, my recommendation is: marginally above the acceptance threshold.
I would be happy to raise my score if the authors could address the points I raised.

---

> ### Author Response · Authors · 2021-11-11
> **Response to Reviewer Ts2w**
>
> We would like to sincerely thank you for your detailed feedback. In particular, we are happy to see that you recognize the contribution of our approach for understanding neural networks via the eigenvectors of the Koopman operator. We are also happy to hear that you find our paper to be well-written and the method to be clearly explained.
>
>
> **Abstract**
>
> We propose to make to following modifications to the abstract:
>
> ... provide limited explanations or depend on local thoeries, such as fixed-point analysis.
>
> ... At the core of our method lies the *Koopman operator*, which is a linear operator that encodes the dominant features of the network latent dynamics. In practice, we compute this operator by representing the hidden states of the network in a basis, and the operator is defined to be the linear fit in that new space. Since it is a linear operator, we can study its eigenvectors and eigenvalues, and we observe they facilitate understanding: ...
>
>
> **Intro**
>
> We propose to make the following modifications to the intro:
>
> ... However, the local nature of these methods is a limiting factor which may lead to inconsistent results. Specifically, their approach is based on fixed-point analysis which allows to study the dynamical system in the neighborhood of a fixed-point (or manifold). In contrast, our approach is global---it does not depend on a set of fixed-points, and it facilitates the exploration of the dynamics near and further away from fixed points.
>
> ... In addition, we demonstrate that the eigenvectors in the ECG classification task naturally identify dominant features in normal beat signals and encode them. Specifically, we show that four Koopman eigenvectors accurately capture the local extrema points of normal beat signals. These extrema points are fundamental in deciding whether a signal is normal or anomalous. Our results reinforce that the network indeed learns a robust representation of normal beat signals.
>
>
> **Section 3**
>
> The typical day-to-day examples are all those cases when deep learning practitioners in industry or academia design and train a model, and they seek to understand the underlying mechanisms of the trained network. For instance, in ECG classification, the practitioner can check if the network correctly identifies the extrema points of normal beat signals using our approach.
>
> moderate size matrix $C$  ->  moderate size matrix approximating the Koopman operator
>
>
> **Section 4**
>
> Importantly, the focus was not on the training and tuning of the networks, but rather on the analysis of the resulting trained networks. Thus, we took the common choices for the hyperparameters and data splits.
>
> For the sentiment analysis, the sizes of the train/test sets are 24690/24713 (no validation set). The hyperparameters we used were reported in Maheswaranathan et al, 2019, and they are reported in Table 1 in the supplemental material.
>
> For the ECG classification, the sizes of the train/test/valid sets are 2481/2226/293. The hyperparameters we used were found using a simple grid search, and we took the model that yields the best average loss value on the validation set. The threshold 26 is set after the network is trained by inspecting the loss values on the validation set.
>
> The ECG network is generative: the model is an autoencoder, trained on normal beat signals. Thus, after training, one can input any signal and inspect its reconstruction. Similarly, one can take any hidden state and pass it through the decoder to receive a signal. Therefore, in contrast to classification models where the network output is vector of pseudo probabilities, the ECG network is an autoencoder which outputs a continuous signal.
>
> We will add below a quantitative comparison to PCA and kernel-PCA which hopefully will be more convincing.

---

### Official Review · Reviewer_XyzC · 2021-11-03

**Correctness:** 2
**Technical Novelty And Significance:** 2
**Empirical Novelty And Significance:** 2
**Recommendation:** 3
**Confidence:** 5

**Main Review:**

Koopman analysis is an interesting technique, and to date, I have not seen it used to try and understand the dynamics of trained neural networks. However, I have a number of concerns about how this paper goes about conducting Koopman analysis, outlined below. I am hoping that the authors can clarify my confusion.

## Major concerns

My understanding of Koopman analysis is that if you want to understand a nonlinear dynamical system, you can find a set of nonlinear, transformed coordinates of the state variables such that there is a _linear_ operator (known as the Koopman operator) that advances the coordinates in the transformed space. This means that we can then use powerful tools from linear dynamical systems analysis to understand the system. The main downside is that we may require an infinite set of transformed coordinates, or that it may be hard to find the right transformation.

Given this understanding, I expected this paper to do something like the following: for a given neural sequence model, find a set of (nonlinear) transformed coordinates such that the nonlinear dynamics of the neural model can be approximated by linear dynamics in the transformed coordinates. Then, study the (linear) dynamics of the transformed coordinates, and show how they relate to the (nonlinear) computations in the original coordinates. If this were the case, I would be very excited about this paper! However, as far as I can tell, this is _not_ what the authors actually did.

Instead, it looks like the authors directly fit a _linear_ dynamical system to the hidden state dynamics. For example, the optimization problem in equations (4) and (5) shows least squares fitting of a _linear_ operator to the hidden states (yes, they use a different linear basis, but it’s still linear, so the whole system $B C B^T$ is still linear as shown in equation (6).

This means that the authors are approximating the nonlinear dynamics with a _global_ linear operator. This strikes me as only being a good idea if the network solves the task using linear dynamics, which may be approximately true for simple tasks, but is most certainly not true for complex tasks. At a minimum, these drawbacks of the method need to be discussed in Section 3. It would be even better to see what happens if one attempts to fit a Koopman operator to a neural network that uses nonlinear dynamics to solve a task. Are the eigenmodes still interpretable? Are they misleading?

For example, if we look at the sentiment classification results, it looks like the network solves the class using a plane attractor, similar to the network analyzed in [Maheswaranathan et al, 2019](https://arxiv.org/abs/1906.10720). (As an aside, I suspect the reason for the plane, as opposed to line, attractor has to do with the amount of l2 regularization applied to the network weights during training. With low values of l2 regularization, networks often learn higher dimensional attractors than necessary. This is also discussed at the end of section 2 in [Aitken et al 2020](https://arxiv.org/abs/2010.15114). At this point, I have to admit the differences between KANN and RENN seem minor. Both identify a linear integration mechanism for solving the task, and both can be used to quantify the “valence” of particular words as seen by the network.

However, I fail to see how the KANN approach is useful at identifying the computations that underlie contextual processing (phrases such as “not bad” or “very good”). The KANN approximation, being purely linear, implies by definition that the network’s interpretation of any word (such as “bad”) is _independent_ of surrounding context. This means that the phrases “not bad and very good” and “very bad and not good” should have the same predictions. Is this the case for the KANN approximation? It might not be true for the underlying (nonlinear) RNN, but must be true for the linear approximation. Is this not a drawback of the KANN approach? By assuming the global dynamics are linear, we hinder our ability to understand how the network processes contextual input.

The analysis done with eigenmode u8 (as an aside, what happened to eigenmodes u5-u7? Why was u8 chosen?) does not really reveal _how_ the network understands the phrase “not bad”. In fact, it does not even tell us what the effect of eigenmode u8 is, because we do not know what the projection of eigenmode u8 is onto the readout weights (output weights) of the network. It’s possible that due to the readout projection, u8 thinks “not bad” has _negative_ sentiment.

I think the analysis in [Maheswaranathan et al 2020](https://arxiv.org/abs/2004.08013) more thoroughly breaks down _how_ the network correctly asses contextual phrases such as “not bad”--through _piecewise_ linear approximations of the dynamics. These piecewise linear approximations are sufficiently nonlinear to describe and understand contextual effects. By contrast, the method proposed here cannot identify these types of computations, because of the assumption of globally linear dynamics.

I am less familiar with the ECG arrhythmia dataset, but I did not quite understand how the pictures in Figure 2 revealed much about the network’s behavior. The reconstruction (which is a linear approximation of the dynamics) will be a linear combination of the projections along each eigenmode, so by definition don’t the eigenmodes have to capture the important features in the signal? That doesn’t seem surprising to me.

## Minor comments

- Section 3.3: Typo: “our C matrices allow _us_ to predict a future state …”
- For any ECG figures, such as Figure 2, what are the units on the time axis, and what is the y-axis?
- In Figure 4, should make it clear in the Figure label that the “percentage” is not accuracy on the test set, but agreement with the baseline model.


**Summary Of The Paper:**

This paper proposes using Koopman analysis to study the dynamics of neural sequence models. The proposal involves fitting linear approximations to the hidden state dynamics of a sequence model.

The paper applies this idea to two applications: RNNs trained on IMDB reviews for sentiment classification, and RNN autoencoders trained on ECG traces used for arrhythmia classification.

**Summary Of The Review:**

This work presents a new application of Koopman analysis for understanding neural sequence models.

However, it does not adequately present drawbacks of the current method (assuming globally linear dynamics), nor does it test the approach in settings where nonlinear computations are required.

---

> ### Author Response · Authors · 2021-11-10
> **Response to Reviewer XyzC**
>
> We appreciate your detailed and thoughtful feedback. We are happy that you recognize the novelty in our paper of using Koopman analysis to understand the dynamics of trained neural networks. We respond to your major concerns below by their order in the review.
>
> **Koopman fitting**
>
> The first paragraph in the major concerns is correct. To better address your next comments, we believe it is better to think of the "transformed coordinates" in terms of the underlying transformation. Namely, to use Koopman, one applies a transformation $\psi$ to the state variables to obtain the observables (transformed coordinates). In the space of observables, the dynamics become linear.
>
> What you describe in your second paragraph is essentially what we are doing---let us explain. The key is the system in question. You mention the neural sequence model and its dyanmics. While our approach can also be used in this context, this is not what we are doing in the paper. The map/dynamics we are interested to study and explain is *the map from inputs to outputs*. For example, let $x_t$ denote the input sequence of words in a review in sentiment analysis and $y_T$ their output label, then the map in question is $\varphi: (x_t)_{t=1}^T \rightarrow y_T$. We use Koopman analysis to study this map and its approximation via a neural network. Thus, the nonlinear transformation $\psi$ which maps the state variables $x_t$ to their transformed coordinates is the network function $F$ that maps inputs to hidden states composed with our choice of basis $B$, i.e., $\psi = B^T \circ F$. This is a nonlinear transformation, and our assumption is that the resulting latent space is "linear enough" to allow Koopman analysis. This is true in practice as the series of papers by Maheswaranathan and co-authors have shown on the sentiment analysis task and other problems. Our analysis shows it is the case for sentiment analysis and ECG classification. Finally, our algorithm is flexible in the choice of $B$, and if the obtained latent space is not linear enough using a linear $B$, one can take a nonlinear $B$ while keeping most of our approach unchanged. In our experiments, a linear $B$ was sufficient, and thus we did not explore a nonlinear option in our current work.
>
> **A global linear operator to represent the network dynamics**
>
> Here we address the comments in the fourth paragraph. In our work, we focus on sentiment analysis and ECG classification. To date, there are at least two full papers that solely deal with the understanding of neural networks for the sentiment analysis problem. In our opinion, this means it is an important task to study and explore. Our work can be seen as a complementary approach to the line of work of Maheswaranathan et al. In addition, it provides insights that are not available elsewhere on sentiment analysis and ECG classification. Importantly, it is not a limitation of our method that we consider a global linear operator, it is simply the case that a global linear operator is good enough for the tasks at hand. Considering our method on a task where the learned latent space is less linear is beyond the scope of this work. To approach such a task, one only needs to compute a nonlinear basis $B$, as we outlined above.

---

> ### Author Response · Authors · 2021-11-10
> **Response to Reviewer XyzC**
>
> **Differences between RENN and KANN**
>
> Here we address the comments in paragraphs five, six, and seven. Indeed, for the sentiment analysis problem, both RENN and KANN successfully identify the integration mechanism and valence of words. However, we believe that our approach is more natural to make this analysis, as the valence and integration directly result from inspecting the eigenvectors and do not require the use of several tools/different matrices as in RENN.
>
> Our approach is not independent of the surrounding context---not by definition, and certainly not in practice. First, as the map from input words to hidden states is given by a recurrent architecture, it holds that every hidden state maintains the context with respect to its past. Second, the premise in Koopman analysis is that the eigenvectors identify dominant structures in the underlying dynamics. In the context of sentiment analysis, these dominant structures are positive/negative 1-grams and positive/negative 2-grams as we show in our results. Third, the predictions obtained with our method achieve very high correspondence (>99%) with the network output as we show in Fig. 4 left panel. This means in particular that our method yields different outputs for phrases as in your example. Specifically, we analyzed in the Sec. 4.1 a batch with 256 samples containing the phrases "not good" and "not bad" and attained >99% corresponding predictions. Given the above, we believe that not only this is not a drawback of our method, but it is one of its key features that we can analyze nonlinear neural mechanisms in a linear fashion.
>
> We used u8 as it highlighted the 2-grams more significantly than other eigenvectors. We have no control over the order of eigenvectors. However, they can be sorted by their dominance by sorting the values in Eq. (10) when summed over time. We can incorporate this into our revision. We will add below results that show the effect of u8 on the readout weights.
>
> **ECG classification**
>
> The key point in the results shown in Fig. 2 is that the eigenvectors capture the important *temporal* features of the signal. This is a key feature of our method and Koopman analysis in general, and typically can not be achieved using other standard approaches such as PCA. We will show an example below comparing the main modes obtained with PCA and kernel-PCA for comparison.
>
> **Minor comments**
>
> We will fix all typos, and improve the labeling in the figures.

---

> > ### Comment · Reviewer_XyzC · 2021-11-19
> > **I still have some confusion about linearity**
> >
> > Hi,
> >
> > Thanks for the response.
> >
> > I am still having trouble understanding how a network with linear state dynamics can give different predictions for "not bad and very good" and "very bad and not good". Let me try and explain how I see the problem:
> >
> > The network processes each token or word at each timestep sequentially. All of the information about previous words seen so far is captured by the hidden state vector at a given timestep. Let's assume the hidden state dynamics are linear (as claimed by the paper). Linearity, by definition, means that order doesn't matter: if the hidden dynamics are linear, it shouldn't matter if the network sees token A and then B, or token B and then A. The effect of each of those inputs sum (because of linearity) and so the output is the same. This means that I can shuffle the sequence of inputs, and the final network response (and therefore the final prediction) will be the same.
> >
> > But if the order doesn't matter, then how does the network correctly process context? How is it that the network's response to "not bad and very good" and "very bad and not good" are the same? (Those two phrases contain the same tokens/words, just in a different order).
> >
> > This is why I don't understand how the KANN analysis reveals how the network processes 2-grams, or any n-grams for that matter.
> >
> > By contrast, Maheswaranthan et al 2020 showed that a _nonlinear_ mechanism is used by the network to process contextual information (it's approximately piecewise linear, but that's still nonlinear!).
> >
> > Perhaps KANN can point us to the subspace in which those nonlinear computations are happening (as indicated by the eigenvector subspace analysis). But, to my understanding, it cannot (or does not) reveal _how_ the network actually performs the nonlinear computation.
> >
> > Can you humor me and make a plot where you take two phrases, shuffle the order, feed them into a linear dynamical system and show that the final states are different?

---

### Official Review · Reviewer_6ZvH · 2021-11-03

**Correctness:** 3
**Technical Novelty And Significance:** 2
**Empirical Novelty And Significance:** 2
**Recommendation:** 5
**Confidence:** 4

**Main Review:**

-The general idea and the results of this work are interesting and can be extended in many directions

-The problem considered is nice, and has a clear audience, but their approach doesn't seem to be novel enough

-The paper is well written, but some parts are not clear enough. Especially, some math parts still need more clarification

*Some issues that, in my view, could improve the paper are*:

- On page 3, section 3.1, line 23, how can we write "h_t ≈ f_t" while two functions h_t and f_t have different of domain and range? In line 24, the function cos(tz) is not a general appropriate example for f_t: M ---> R (it is only true for f_t: R---> R)

- In section 3.2 some notations and explanations are not so clear. In particular, there is no discussion on the size of the matrices B and C

- There are some grammatical mistakes as well as typos

**Summary Of The Paper:**

The paper proposes a framework for studying sequence neural models based on Koopman theory. Particularly, the authors compute linear approximations of the state paths via simple matrix-vector multiplications. Also, the dominant features of the dynamical system and their effect on inference and prediction is determined. Their results on the sentiment analysis problem, and the ECG classification challenge provide simple yet precise descriptions of the underlying dynamics and behavior of the recurrent models.

**Summary Of The Review:**

Some parts of the paper are not so clear and require more clarification. Their approach doesn't seem to be novel enough. The experimental results and comparisons to other existing methods are inadequate.

---

> ### Author Response · Authors · 2021-11-10
> **Response to Reviewer 6ZvH**
>
> We appreciate your feedback, and that you recognize that the idea and results are interesting. As we clearly specify in the paper, the tool (applied Koopman operator) we use is not new. The main contributions in our paper are the application of Koopman methods in the context of understanding sequence neural models, and the analysis on sentiment analysis and ECG classification. To the best of our knowledge, this has not been done before. We would appreciate to receive feedback regarding the novel contributions of our paper, and we anticipate your response. In addition, we are happy to improve the mathematical parts that need more clarification. Could you please specify what are those parts? We will attach below a paragraph explaining how $h_t$ and $f_t$ can be associated. The dimensions of $B$ is $k \times r$, where $k$ is the hidden dimension of the network and $r\leq k$. Thus, the dimensions of $C$ are $r \times r$. We will fix all grammatical mistakes and typos in the revised version. Finally, can you elaborate on the inadequacies that you find in the experimental results and comparisons?

---

> ### Author Response · Authors · 2021-11-10
> **Approximating $f_t$ with $h_t$**
>
> The example in the paper discussing $h_t$ and $f_t=cos(tz)$ was only to gain intuition. We are happy to include the following detailed explanation in the revised version. Given any scalar function $f_t:M \rightarrow \mathbb{R}$, one can approximate it using a vector $h_t \in \mathbb{R}^k$ in the following sense.
> 1. Sample $k$ points from the domain $M$, and denote them by $x_j$ for $j=1,...,k$.
> 2. define $h_t(j) = f_t(x_j)$ for every $j$.
>
> That is, $h_t$ approximates $f_t$ by sampling $f_t$ at a fixed predefined set of points.
>
> There are many ways to extend the example in Line 24 to describe the more general case of functions $f:M \rightarrow \mathbb{R}$. For instance, let $f_t$ be the function $f_t(z) = |(cos(tz(j)))|_2$ where $z \in \mathbb{R}^m$, and $z(j)$ is the $j$-th component of $z$. That is, we compute the function $cos(ty)$ on every coordinate of $z$, and stack the result in a vector and take its norm.

---

### Decision · Program_Chairs · 2022-01-20

**Decision:**

Reject

**Comment:**

This paper proposes to apply the Koopman operator theory framework for analyzing sequence neural models. The authors considered two particular applications, namely sentiment analysis and ECG (electrocardiogram) classification.

Reviewers generally agree that the results obtained on the two tasks are interesting. However, there are concerns that the paper lacks methodological novelty (concerning the Koopman operator framework, which the authors agreed) and that the paper would be more suited for an applied conference and/or journal.